# Hybrid Nanogel Drug Delivery Systems: Transforming the Tumor Microenvironment through Tumor Tissue Editing

**DOI:** 10.3390/cells13110908

**Published:** 2024-05-24

**Authors:** Theodora Katopodi, Savvas Petanidis, George Floros, Konstantinos Porpodis, Christoforos Kosmidis

**Affiliations:** 1Laboratory of Medical Biology and Genetics, Department of Medicine, Aristotle University of Thessaloniki, 54124 Thessaloniki, Greece; katopodi@auth.gr; 2Department of Pulmonology, I.M. Sechenov First Moscow State Medical University, Moscow 119992, Russia; 3Department of Electrical and Computer Engineering, University of Thessaly, 38334 Volos, Greece; gefloros@e-ce.uth.gr; 4Pulmonary Department-Oncology Unit, G. Papanikolaou General Hospital, Aristotle University of Thessaloniki, 57010 Thessaloniki, Greece; kporpodis@auth.gr; 5Third Department of Surgery, AHEPA University Hospital, Aristotle University of Thessaloniki, 55236 Thessaloniki, Greece; dr.ckosmidis@gmail.com

**Keywords:** nanogels, immunotherapy, reprogramming, tumor editing

## Abstract

The future of drug delivery offers immense potential for the creation of nanoplatforms based on nanogels. Nanogels present a significant possibility for pharmaceutical advancements because of their excellent stability and effective drug-loading capability for both hydrophobic and hydrophilic agents. As multifunctional systems, composite nanogels demonstrate the capacity to carry genes, drugs, and diagnostic agents while offering a perfect platform for theranostic multimodal applications. Nanogels can achieve diverse responsiveness and enable the stimuli-responsive release of chemo-/immunotherapy drugs and thus reprogramming cells within the TME in order to inhibit tumor proliferation, progression, and metastasis. In order to achieve active targeting and boost drug accumulation at target sites, particular ligands can be added to nanogels to improve the therapeutic outcomes and enhance the precision of cancer therapy. Modern “immune-specific” nanogels also have extra sophisticated tumor tissue-editing properties. Consequently, the introduction of a multifunctional nanogel-based drug delivery system improves the targeted distribution of immunotherapy drugs and combinational therapeutic treatments, thereby increasing the effectiveness of tumor therapy.

## 1. Introduction

Nanocarriers have grown in prominence in biomedicine as a result of the development of advanced nanotechnology in recent decades [1,2]. Due to their ability to encapsulate drugs, nanocarriers are not only used as delivery systems for standard chemotherapeutic agents but also as platforms for combination therapy, multifunctional diagnostics, and theranostics [3,4]. As a primary adaptable drug delivery system (DDS), nanocarriers have been used for a variety of disease therapies. These include passive targeting due to the EPR effect, active targeting by ligand modification of nanoplatform surfaces, and site-specific and time-controlled drug delivery strategies mediated by stimuli-responsive materials [5]. Nanogels are a type of systemic drug delivery nanocarrier. Specifically, nanogels are hydrogels with a 3D tunable porous structure and a particle size ranging from 20 to 280 nm; they can be distinguished from microgels and in situ-forming hydrogels, which enable local delivery [6,7]. Nanogels are made up of different natural polymers, synthetic polymers, or mixtures of both, which helps to encapsulate proteins, oligonucleotides, and tiny compounds [8]. Nanogels can be utilized for imaging, diagnosis, and medication delivery because of their unique and distinctive properties [9]. Nanogels, as a special type of hydrogel, can maintain a highly hydrated state and shrinking-swelling characteristics in a variety of environmental settings [10]. Their 3D structure makes it possible to encapsulate hydrophobic or hydrophilic agents in their internal network, potentially shielding these compounds from deterioration during storage or in circulation (hydrolysis or enzymolysis) [11]. Nanogels have effective controlled drug release features because, in contrast to conventional nanoparticles (NPs), they have variable particle sizes and shapes as well as sensitivity to external stimuli such pH, temperature, ionic strength, and redox conditions [12]. In addition, surface modification can increase the circulation time of nanogels and modify their multifunctionality and targeting [13]. Nanogel-based DDSs have gained popularity and made a significant effect in recent years due to the aforementioned benefits [14]. The most recent advancements in nanogels and their biological use in drug administration are discussed in this review. Furthermore, the production of several pH/temperature/redox-responsive nanogels is analyzed in detail, in addition to the composition of nanogels and their synthesis techniques.

## 2. Bioengineered Nanogels

Chemical crosslinking and physical self-assembly are two types of nanogel synthesis methods that can be categorized according to the many architectures and building blocks of the resulting nanogels [15]. By using covalent crosslinks between functional units on polymer chains, nanogels created by chemical crosslinking display better stability compared to nanogels created by physical crosslinking [16]. Meanwhile, non-covalent interactions—which mostly consist of hydrogen bonds, Van der Waals forces, hydrophobic interactions, host–guest interactions and electrostatic interactions—are frequently responsible for the reversible connections of physically crosslinked nanogels [17]. Numerous benefits of these crosslinked nanogels have been demonstrated in the realm of cancer imaging. With better stability than micelles and liposomes, crosslinked nanogels can limit burst release during circulation, enhancing tumor accumulation and drug delivery effectiveness [18,19]. Because they are structurally stable and have three-dimensional cross-linking, nanogels can easily withstand shear forces and serum proteins in the bloodstream [20]. Additionally, because of the longer circulation life, there is a greater likelihood that nanogels will accumulate in the tumor, resulting in deep tumor penetration. By adding stimuli-responsive functional groups, nanogels can target particular immune cell subpopulations (like Tregs, MDSCs, macrophages, dendritic cells, etc.) or immune organs, increasing the drug bioavailability and inhibiting immune-related adverse effects and chemoresistance. Furthermore, since there is an increased surface area, off-target impacts may be minimized or eliminated [21,22]. The host’s immune system is gradually awakened by the slow release once it reaches the tumor location, leading to improved therapeutic benefits.

## 3. Nanogels in Biomedicine

Due to their stimulus-responsive characteristics, nanogels are used to treat numerous medical disorders, including cancer, neurological disorders, and cardio-inflammatory conditions [23,24]. Body functionality changes under various pathological circumstances (cancer or inflammation) as a result of altered metabolic and/or physiological states. It is extremely difficult to demonstrate a correct drug release profile and therapeutic effects using current delivery systems since they are unable to react to physiological variables like pH and temperature [25,26]. In such circumstances, nanogels are highly helpful since their stimulus responsiveness multiplies several times, enabling the release of the necessary medication to the chosen therapeutic site [27]. The reduction intoxicity caused by the transdermal distribution of active pharmaceutical ingredients, such as aceclofenac-loaded NGs, is one of the numerous factors for choosing NGs as a drug delivery system in clinical trials [28]. Hybrid nanogels in clinical trials are predicted to selectively kill tumor cells via multifunctional strategies without hurting normal tissues because of their effective drug release, responsiveness, high tumor targeting efficiency, and multifunctionality. These engineered carriers can also inhibit cancer-related chemoresistance and tumor relapse associated with cancer therapy [29,30].

## 4. Hybrid Nanogels

Hybrid nanogels can react to both internal and external stimuli [31]. Inorganic nanomaterials and organic nanomaterials (carbon-based) have both been added into nanogels to create multipurpose, highly responsive, hybrid nanocarriers [8]. Hybrid nanogels, as opposed to bare nanoparticles, have the softness and fluidity that are typically associated with nanogels. This is extremely advantageous for drug delivery systems since it makes it easier for cells to absorb the nanocomposite [32,33]. The extended circulation time exhibited by nanogels is another benefit offered by hybrid nanogels but not by bare nanomaterials. In fact, the longer the nanomaterial stays in circulation, the simpler it is for it to elude the reticuloendothelial system and the softer and larger the nanomaterial is, as was shown inhuman A549 lung carcinoma-bearing mice [34]. Several drug compounds with distinct physicochemical properties can be delivered together because of the special hybrid and compartmentalized architectures of nanogels [35,36]. For example, DOX/ICG coencapsulated liposome-coated thermosensitive nanogels can be used for NIR-photothermal synergistic therapy for 4T1 breast cancer cells (Figure 1). Furthermore, imaging techniques for diagnosing diseases use hybrid nanogels that contain optical and magnetic resonance imaging contrast in 4T1-tumor bearing mice [37]. More crucially, the stimuli-responsive properties can be easily included into the design of targeted drug release strategies through nanogel-based multifunctional drug delivery systems [38]. Various hybrid nanogels and multicompartment nanogels are analyzed in this article. The use of designing stimuli-responsive multifunctional nanogels as drug delivery systems, and the co-delivery of various agents and imaging agents for diagnosis and theranostics is really promising for the future of modern pharmaceutics.

## 5. Nanogels for Cancer Immunotherapy

Immunotherapy, which primarily focuses on immune system activation and improved anti-tumor immunity, has grown to become the standard approach for treating cancer in recent years [40]. Immunotherapeutic techniques have shown encouraging results, but their applicability is constrained by immune-related side effects and poor immune response rates [41]. Due to their structural stability and ability to exhibit properties similar to natural tissues, nanogels are aqueous materials that have immune-editing properties. As a result, they are able to withstand shear forces and serum proteins in the bloodstream [42]. Additionally, because of their longer circulation life, nanogels are more likely to accumulate in tumors, providing deep tumor penetration. The large specific surface area can be used to introduce stimuli-responsive functional groups, allowing for various physical and chemical modifications to improve targeting to particular immune cell subpopulations or immune organs, increase the drug’s bioavailability, and confer a low risk of immune-related adverse events in nanogel therapies [43]. Consequently, injectable nanogels or hydrogels can be employed as effective local delivery systems to greatly enhance the induction of antitumor immune responses and increase the effectiveness of cancer immunotherapy [44]. Specifically, when combined with other traditional cancer therapies including chemotherapy, phototherapy, and radiotherapy, injectable hydrogels can quickly and effectively enhance the activity of anticancer drugs by increasing the therapeutic efficacy [45]. In this scenario, damage-associated molecular patterns (DAMPs) are produced in cancer cells by locally injected therapeutic chemicals that might cause immunogenic cell death (ICD) through particular therapeutic regimes, such as photodynamic therapy (PDT) and photothermal therapy (PTT) [46,47]. When antigen-presenting cells (APCs) phagocytose tumor-associated antigens, innate immune responses against cancer cells are subsequently activated on their own. Additionally, when paired with immunological modulators or adjuvants, such as immune checkpoint blockades (ICBs) and Toll-like receptor agonists, these substances can enhance the cancer-immunity cycle, resulting in the development of powerful T cell-mediated adaptive immunity [48]. Furthermore, combining cystamine with carboxymethyl chitosan (CMCS)-originated polymetformin creates a stimuli-responsive nanogel (PMNG) through combinational chemo-immunotherapy. Specifically, the chemotherapy drug doxorubicin (DOX) was included into PMNG, and a hyaluronic acid coating was added to enhance the nanogel’s overall biocompatibility and targeting capabilities (D@HPMNG). PMNG notably recruited intratumoral CD8+ T cells and reprogrammed tumor-associated macrophage phenotypes, reshaping the tumor immune microenvironment. As a result, following surgical resection, D@HPMNG therapy significantly reduced melanoma development and prevented its recurrence [49]. In a similar manner, the CCL21a/ExoGM-CSF+Ce6 @nanoGel was created by combining CCL21a and ExoGM-CSF+Ce6 (tumor cell-derived exosomes with granulocyte–macrophage colony-stimulating factor (GM-CSF) mRNA encapsulated inside and sonosensitizer chlorin e6 (Ce6) incorporated in the surface). The time-released CCL21a and GM-CSF are released by the modified hydrogel. This modular nanogel vaccine, with its two programmable modules, effectively impeded tumor development and metastasis, eliminating the trapped tumor cells, and triggering potent and extended immunotherapy in a coordinated fashion [50]. Furthermore, for TME-responsive drug release, self-degradable PMI nanogels containing imiquimod (Imi) and metformin (Met), two repurposed immune modulators, were designed. These nanogels modify the TME in the following ways: encouraging the development of dendritic cells, repolarizing tumor-associated M2-like macrophages, and reducing the expression of PD-L1. In the end, PMI nanogels effectively encouraged CD8+ T cell invasion and activation while reshaping the immunosuppressive TME [51]. Overall, these findings provide credence to the idea that nanogels may be a useful combination therapeutic regime for boosting current cancer immunotherapy approaches.

## 6. Injectable Reprogrammable Nanogels/Hydrogels

Injectable reprogrammable nanogels can be injected into tumor sites to reprogram or edit tumor and immune cells inside the TME [52,53]. A variety of treatments, including as chemo/immunotherapy drugs, antibodies, proteins, and nucleic acids, can be easily packed using injectable hydrogels during the gelation processes. Additionally, these therapies can retain a relatively high drug concentration with reduced systemic toxicity by sustainably releasing the drug from the nanogels into the tumor location [54]. Specifically, the creation of size-controllable DNA nanogels from the self-assembly of DNA nanostructures through multivalent host–guest interactions highlights the potential of reprogrammable DNA nanogels in drug delivery (Figure 2) [55]. In comparison to physically crosslinked hydrogels, chemically crosslinked hydrogels often offer stronger mechanical characteristics, a lower critical gelation concentration, and a slower rate of breakdown [56]. Hydrogels can be created by chemically crosslinking a variety of polymers. A characteristic example is the F127/PEG hydrogel that is created using non-swellable di-acrylated Pluronic F127 (F127-DA) to create a hydrogel with microwells for the production of vascular spheroids that demonstrated more mechanical strength than a typical di-acrylated polyethylene glycol (PEG-DA) hydrogel. Human umbilical vein endothelial cells (HUVECs) and fibroblasts created uniformly sized vascular spheroids on their own in the microwells. Nitric oxide (NO), prostacyclin (PGI2), and tissue factor pathway inhibitor (TFPI) secretion demonstrated that the endothelial functions of vascular spheroids were one-fold higher than those in a two-dimensional (2D) culture. It is interesting to note that vascular spheroids with larger diameters show more sensitivity to ethanol toxicity than smaller-diameter spheroids and this is because large spheroids have stronger endothelial functions [57].

Recently, in glioblastoma treatment, the development of the membrane-coated nucleic acid nanogel Vir-Gel, which is embedded with therapeutic miRNA, can change the pro-invasive M2 phenotype of macrophages and microglia into the anti-tumor M1 phenotype. The miR155-bearing nucleic acid nanogel’s targetability and cell absorption efficiency are greatly improved by Vir-Gel by simulating the virus infection process. According to in vivo tests, Vir-Gel extends the lifetime of miR155 in circulation and gives it active tumor-targeting abilities and outstanding tumor inhibitory efficiency. The virus-mimicking nucleic acid nanogel reprograms microglia and macrophages for efficient glioblastoma therapy [59]. Furthermore, injectable hydrogels made of graphene oxide (GO) and polyethylenimine (PEI), which, for at least 30 days after subcutaneous injection, can produce mRNA (for ovalbumin, a model antigen) and R848-loaded nanovaccines after administration. The dispersed nanovaccines have the ability to deliver specifically to lymph nodes and shield mRNA against deterioration. The data indicate that with just one treatment, this transformable hydrogel can greatly boost the number of antigen-specific CD8+ T cells and consequently suppress tumor growth. In the interim, this in situ transforming RNA hydrogel/nanovaccine can cause the serum to produce an antigen-specific antibody, which in turn stops metastasis from occurring [60]. Likewise, in osteosarcoma, a sequential nanocomposite hydrogel was created that alters the behavior of CD45+ T lymphocytes and promotes their infiltration into tumors. The method uses an injectable combination of tetrabasic polyethylene glycol and carboxymethyl chitosan to generate a hydrogel that allows for the controlled release of liposomal doxorubicin (L-Dox) and a powerful CAF suppressor (Nox4 inhibitor, Nox4i) to cause immunogenic cell death (ICD) when administered in situ. In stroma-rich osteosarcoma models, Nox4i efficiently inhibited CAF activation, circumventing T-cell exclusion mechanisms and allowing for the release of L-Dox on schedule for ICD induction. The efficacy of the co-delivery gel was significantly increased when combined with the αPD-1 checkpoint inhibitor and substantially boosted tumor T-cell infiltration and favorable anti-tumor immunity [61].

## 7. Nanogel-Induced Photoimmunotherapy

Reprogrammable nanogels can also be used for photothermal (PTT) and photodynamic (PDT) therapy as potential therapy regimens for tumor ablation and necrosis. Eliminating solid tumors is the ultimate objective of photoimmunotherapy based on nanogels, such as PTT, which produces heat, and PDT, which produces reactive oxygen species (ROS) and also triggers a range of anti-tumor immune responses. This approach has attracted a lot of interest and research in recent years due to its great clinical therapy success with low invasion and weak adverse effects. Recently, Ding et al. developed an siRNA-mediated low-temperature PTT therapeutic combination using a polydopamine (PDA)-coated nucleic acid nanogel (Figure 3) [62]. In detail, DNA-grafted polycaprolactone (DNA-*g*-PCL) forms into nanosized hydrogel particles through nucleic acid hybridization; siRNAs target the heat-shock protein 70 (Hsp70) and act as crosslinkers. The generated siRNA-embedded nanogels are then further covered with a thin coating of polydopamine, which gives the nanogels an exceptional photothermal conversion capability when exposed to NIR light irradiation in addition to protecting them against enzymatic degradation. This triple-shielded siRNA delivery complex exhibited the ability to effectively ablate the tumor under acidic conditions following surface PEGylation. In addition, a cisplatin-containing nanogel functionalized with photothermal gold nanorods (GNRs) that are electrostatically coated with doxorubicin (DOX) was reported as an anticancer drug delivery method. Hyaluronic acid and the auxiliary anticarcinogen CDDP crosslink to generate the nanoparticles in the presence of DOX-decorated GNRs. The resultant biomimetic nanocarrier (4T1-HANG-GNR-DC) displays effective accumulation through homologous tumor targeting and possesses long-term retention in the tumor microenvironment since the nanogel is covered with a cancer cell membrane. To produce a synergistic photothermal/chemotherapy, NIR laser irradiation causes in situ photothermal therapy, which further generates hyperthermia-triggered on-demand drug release from the nanogel reservoir [63]. In a similar manner, a tumor-targeted injectable double-network hydrogel was used to prevent breast cancer recurrence and wound infection using synergistic photothermal therapy with brachytherapy. By inhibiting the self-repair of damaged DNA and enhancing blood circulation to alleviate the hypoxic microenvironment, hyperthermia brought on by photothermal therapy can synergistically boost the therapeutic efficiency of brachytherapy and simultaneously eradicate pathogenic microorganisms. Due to the isotope labeling of the loaded 125 I-GNR-RGDY, this nanocomposite hydrogel promotes antibacterial activity to prevent potential wound infection and can be tracked by single-photon emission computed tomography imaging [64]. Photothermal therapy can also be used synergistically with siRNA therapy. For example, PD-L1 siRNA and a photosensitizer can be co-packaged in a nucleic acid nanogel for synergistic cancer photoimmunotherapy. In this approach, phosphorothioate modification sites for pheophorbide A (PPA) photosensitizers are effectively grafted onto DNA. The four PPA-grafted DNAs that were created are arranged into a tetrahedron framework, which joins with a PD-L1 siRNA by supramolecular self-assembly to create a siRNA and PPA co-packaged nanogel. The nanogel can photodynamically kill tumor cells and cause marked immunogenic cell death since it contains two therapeutic chemicals. Additionally, it simultaneously inhibits the tumor cells’ PD-L1 expression, which significantly boosts the antitumor immune response and results in improved antitumor efficacy in a synergistic manner [65].

## 8. Engineered Nanogels for Tumor Tissue Editing

In order to achieve tumor control or resolution, current tumor treatment approaches attempt to introduce biological “editing hallmarks” in tumor tissues [66,67]. In metastatic cancer, multi-drug strategies aim to re-edit tumor tissues by ‘editing’ heterotypic cell types inside the TME, which are responsible for tumor cell differentiation, cancer metabolism, blockage of immunosurveillance, and chemoresistance [68].

For the purpose of editing tumor tissues, a wide range of reprogramming methods using nanogel drug delivery systems have been employed [69,70]. For example, an engineered non-cationic DNA-crosslinked nanogel for Cas9 and a single guide RNA (Cas9/sgRNA) complex intracellular delivery was designed. The Cas9/sgRNA complex is first loaded onto a DNA-grafted polycaprolactone brush (DNA-*g*-PCL), which is then crosslinked by DNA linkers using nucleic acid hybridization to create a nanosized hydrogel in which the gene editing tools are embedded and safeguarded. Its compact construction, superior physiological stability against nuclease digestion, and improved cellular absorption efficiency make the Cas9/sgRNA complex-containing nanogel a viable tool for targeted genome editing (Figure 4) [71]. Furthermore, the homologous delivery of the chemotherapeutic doxorubicin (DOX) to breast cancer was achieved using CRISPR/Cas9-edited Pd-l1KO TDEV-fusogenic anthracycline liposomes with a high drug encapsulation efficiency (97%) to enhance the immunogenic response and promote PD-L1 overexpression in the tumor. In the 4T1-bearing TNBC mouse model, sequential administration of disulfide-linked PD1-cross-anchored TDEV nanogels at one-day intervals could sustainably release PD1 in the tumor, leading to a high percentage of effector T cell-mediated destruction of orthotopic and metastatic tumors without off-target side effects. A solid base for creating chemoimmunotherapeutic formulations for TNBC therapy at the clinical level is provided by this effective cancer-homing delivery capacity and TDEV-tandem-augmented chemoimmunotherapeutic technique [72]. In addition, a “tumor-editing hydrogel” loaded with light-controlled immunomodulatory engineered cells (FLICs) allows for the precise control of the release of cytokines (IFN-, TNF-, and IL-12) in response to illumination. Studies using a mouse model of B16F10 melanoma resection revealed that immunomodulatory cytokines can be sustainably released from FLIC-loaded hydrogel implants put at the surgical wound site, preventing tumor recurrence and extending animal survival. Additionally, the hydrogel implants loaded with FLICs induced long-term immunological memory that guarded against tumor recurrence. The results showed that optogenetic perioperative immunotherapy using hydrogel implants loaded with FLICs provides a secure therapeutic alternative for solid tumors by inducing the host’s innate and adaptive immune systems to prevent tumor recurrence after surgery [73].

## 9. Re-Editing the Immunosuppressive Tumor Microenvironment

The immunosuppressive tumor microenvironment plays a key role in tumor progression and metastasis [74]. The extracellular matrix (ECM), extracellular matrix cells, immune cells, and numerous transcription factors that make up the TME all contribute to the growth of the tumor [75]. Under the attraction of chemokines and cytokines in the TME, a variety of immunosuppressive immune cells infiltrate into the tumor, resulting in a significant level of immunosuppression [76]. Engineered nanogels can edit/alter the TME’s special properties (acidic pH, hypoxia, altered metabolism), the interactions of immune cells with tumor cells, and the secretion of immunosuppressive substances, which are all factors that contribute to maintaining an immunosuppressive microenvironment and causing immune tolerance, which significantly reduces the efficacy of cancer vaccine-induced immunotherapy [77,78].

Specifically, hollow inorganic/organic hybrid nanogels can serve as a multimodal theranostic vehicles for the guidable delivery of MRI diagnostic imaging and hyperthermia/chemotherapy strategies (Figure 5). In addition, Xu et al. reported a GSH/ROS dual response nanogel system (IM) that can actively target cancer cells’overexpression of mannose receptors (MRs), utilize indocyanine green’s (ICG) ultra-stable photothermal capacity, trigger cell pyroptosis, and improve tumor immune responses. Through photoactivated ICG, photo-triggered IMs cause cytoplasmic Ca^2+^ introgression and activate caspase-3. Disconnecting Se-Se links has the potential to disturb the equilibrium between oxidation and reduction in tumor cells, leading to oxidative stress and caspase-3 cleavage that is ultimately responsible for controlling cell pyroptosis. When used in conjunction with anti-programmed death receptor 1 (anti-PD-1), this nanogel technology effectively suppresses both local and distant tumors while also extending mouse longevity [79]. The poor response rate of existing cancer immunotherapies points to the TME’s abundance of immunosuppressive elements and dearth of antigen-specific T cells. For that reason, an immunomodulatory multidomain nanogel (iGel) that can be injected and overcome the restriction by reprograming the pro-tumorigenic TME to activate antitumorigenic immune niches. Immunosuppressive cells are reduced as a result of the local and prolonged release of immunomodulatory medicines from the iGel, which also cause immunogenic cell death and enhanced immunogenicity. Systemic antitumor immunity and a memory T cell response are produced when the iGel is used as a local postsurgical treatment, and the recurrence and metastasis of malignancies to the lungs and other organs are considerably reduced. Non-responding checkpoint blockade therapy groups are converted into responding groups through TME reshaping with the iGel. Overall, this iGel nanovaccine is anticipated to serve as an immunotherapeutic platform with less systemic toxicity that can remodel immunosuppressive TMEs and synergize cancer immunotherapy with checkpoint treatments [80].

Combined antitumor effects of chemotherapy and immunotherapy can also be achieved with a biomimetic nanogel with tumor microenvironment responsiveness (Figure 6). In a similar manner, hybrid nanogels use two oppositely charged chitosan derivatives and hydroxypropyl-cyclodextrin acrylate for paclitaxel binding and precisely regulating pH responsiveness, respectively. Anerythrocyte membrane supported by nanogel can accomplish “nanosponge” interleukin-2 delivery without losing its bioactivity. These nanogels dramatically elevated the anticancer efficacy through improved drug penetration, induction of calreticulin exposure, and increased antitumor immunity by responsively releasing the drugs in the TME. The combination of these medications at low doses alters the tumor microenvironment, as shown by the accelerated infiltration of immune effector cells and the decline in immunosuppressive factors [82].

## 10. Reprogrammable Nanogels for Tumor Hypoxia

The quantity of reactive oxygen species (ROS) produced by mitochondrial metabolic activities can also be increased by tumor hypoxia [84]. As signaling molecules that influence numerous physiological processes including DNA mutagenesis, protein inhibition or activation, protein apoptosis, and inflammation, ROS play a key role inhypoxic tumorigenesis [85]. High levels of hypoxia across the microenvironment limit immune responses via regulatory T cell-mediated tumor immunosuppression, despite the fact that locally high concentrations of ROS are employed to kill tumor cells in tumor PDT therapy [86,87]. Furthermore, in metastatic cancer, hypoxia is crucial for the establishment of chemoresistance, immunosuppression, and tumor progression via extravasation [88]. To address this issue, a gelatin and ferulic acid hydrogel that can construct hydrogel networks by consuming oxygen in a laccase-mediated reaction was synthesized. This hydrogel is hypoxia-inducible (HI), and it can also be used to create other hydrogels. The hydrogels’ oxygen gradients and levels may be precisely regulated and anticipated. These HI hydrogels can facilitate rapid neovascularization from the host tissue during subcutaneous wound healing and direct vascular morphogenesis in vitro through the activation of matrix metalloproteinases by hypoxia-inducible factors. These engineered HI-hydrogels can be used in a wide range of diseases, from cancer, hypoxia-induced inflammation, and hypoxia-related disorders like metabolic syndrome and diabetes [89]. In addition, hypoxia can induce mitochondrial dysfunction and T cell exhaustion, which is a major obstacle in T cell-based tumor immunotherapy. To solve this, an injectable hydrogel was created to simultaneously coordinate MHC I expression and T cell exhaustion for enhanced cancer immunotherapy [90]. Axitinib was encapsulated in the lipid bilayer of the tumor cell membrane vesicle (O-TMV), while 4-1BB antibody and proprotein kexin type 9 inhibitor PF-06446846 nanoparticles were present in the cavities of the hydrogel. Axitinib’s reduction of hypoxia and the enhanced T cell mitochondrial biogenesis work together to cure T cell exhaustion, while PF-06446846′s upregulation of MHC I expression triggers T cells to recognize tumor cells, suggesting a potent immunotherapeutic effect. This novel approach of reversing T cell exhaustion and enhancing T cell potency presents a novel idea for T cell-based cancer immunotherapy. Likewise, the creation of a hypoxia-degradable zwitterionic poly(phosphorylcholine)-based (^H^PMPC) nanogel for tumor drug delivery and the manufacture of a novel hypoxia-responsive crosslinker demonstrated lengthy blood circulation and favorable immunological compatibility, which caused significant and persistent accumulation in tumor tissue. The ^H^PMPC nanogel induced a stable immune response and desirable immune compatibility, which led to high and long-lasting accumulation in tumor tissue. Due to the thorough drug release, the ^H^PMPC nanogel showed superior tumor suppression efficacy both in vitro and in vivo compared to a reduction-responsive phosphorylcholine-based nanogel (Figure 7) [91].

## 11. Conclusions

Biomedical nanogels have recently made significant clinical progress thanks to the quick development of nano/biomaterials, and many of them have been commercialized for a variety of uses, such as in antibacterial coatings and scaffold tissue engineering, demonstrating the significant clinical potential of nanogels as a delivery scheme [92,93]. However, nanogel-based cancer vaccines have not yet received clinical approval, possibly as a result of technical challenges with the preparation and storage, implantation, and injection at tumor target areas [94]. Despite the fact that nanogel-based cancer vaccines have been shown to encourage immune cell recruitment, reduce immunosuppression, and elicit a potent tumor-specific immune response, there are still numerous challenges to be addressed [95]. Nevertheless, shape memory hydrogels and injectable self-healing nanogels have demonstrated rapid development in recent years. The development of hydrogel-based cancer vaccines will still require more creative methodologies and preparation techniques [96]. Numerous nanogel systems have been investigated and found to elicit a potent immune response in addition to hydrogel-based cancer vaccines [97]. The creation of nanogels has been considerably aided by a number of useful nano/biomaterials, including polymers, inorganic nanoparticles, and liposomes [98]. Nanogels can be loaded with numerous components and shield them against deterioration, much like hydrogel-based cancer vaccines. There are numerous immunogenic nanogel carriers that can be utilized as immunoadjuvants to boost immune responses [99]. Additionally, surface alterations and size adjustments can improve targeting even more.

In order to obtain the optimal tumor immunotherapy approach based on the patient’s immune system and TME, unique and improved designs of injectable nanogels are required [100]. Due to their capacity to incorporate and combine a wide range of the desired characteristics for drug delivery and cancer therapy, multifunctional stimuli-responsive hybrid nanogels appear to hold enormous potential in the field of oncology (Table 1) [101]. Future research could focus on the in vivo compatibility of nanogels, their capacity for overcoming tumor heterogeneity, their potential toxicity, and the impact of endocytic entrapment on hybrid nanogels [102,103]. To make this clinical transition easier, non-therapeutic elements including regulatory guidelines, standardized assays, appropriate animal models, and good manufacturing practice (GMP) also need to be taken care of. For this reason, we should make injectable nanogels that can reactivate the immune system’s ability to fight cancer cell expansion as well as to deliver therapeutic tumor agents directly into chemoresistant immunosuppressive tumors. By incorporating immunogenic characteristics into the breakdown products of injectable nanogels, immunological responses can be constantly stimulated by self- or external stimulus-based disintegration. Additionally, by developing methods to combine immunotherapy with therapeutic modalities and enhancing their physicochemical qualities and biosafety, nanogels have the potential to significantly speed up the clinical translation.

## Figures and Tables

**Figure 1 cells-13-00908-f001:**
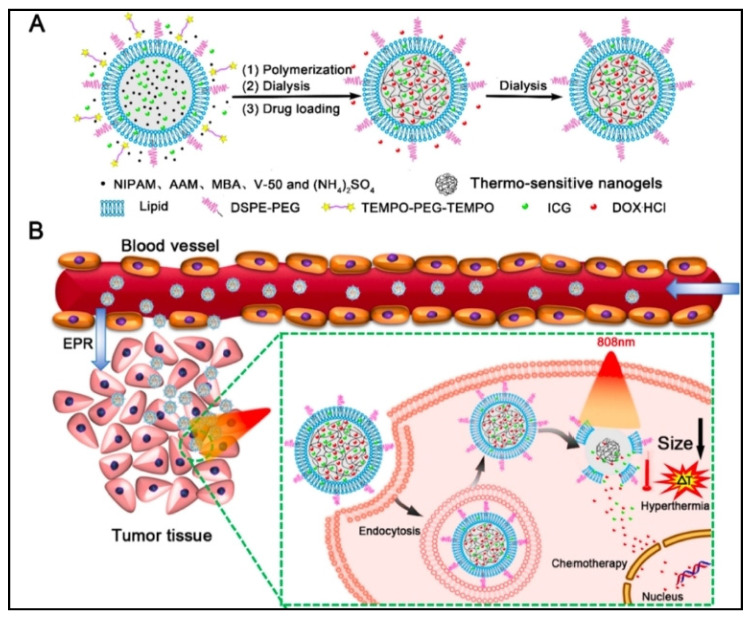
DOX/ICG liposome-coated thermosensitive nanogel used for NIR-photothermal synergistic therapy. (**A**) Liposome-coated poly (N-isopropylacrylamide-co-acrylamide) (P(NIPAM-co-AAM)) nanogels loaded with the NIR dye indocyanine green (ICG) and DOX were created by employing liposomes as a template for in situ polymerization. (**B**) NIR-driven hyperthermia-stimulated intracellular DOX release (b). Reproduced with permission from Ref. [39]. Copyright 2018, ACS Publications.

**Figure 2 cells-13-00908-f002:**
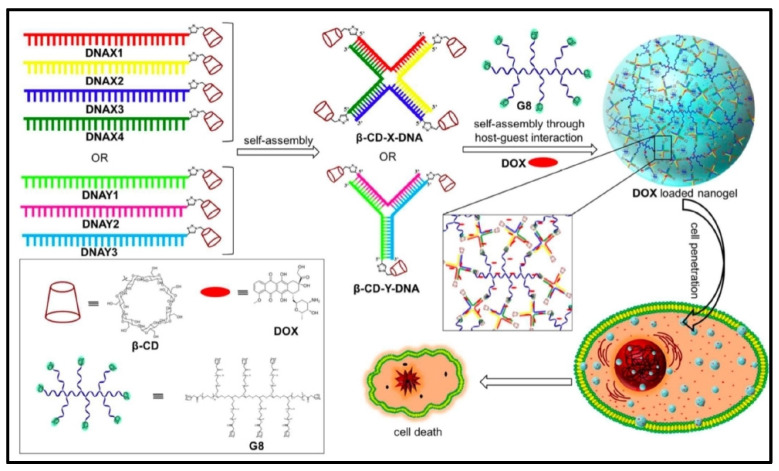
Controllable DNA nanogels for drug delivery. Graphic scheme of DNAX1-4, DNAY1-3, and their self-assembly into beta cyclodextrin (β-CD). In the presence of DOX, the self-assembly of β-CD-coupled XDNA/YDNA with G8 results in the creation of composite DNA nanogels that are loaded with DOX and are suitable for drug administration applications. Reproduced with permission from Ref [58]. Copyright 2017, Royal Society of Chemistry.

**Figure 3 cells-13-00908-f003:**
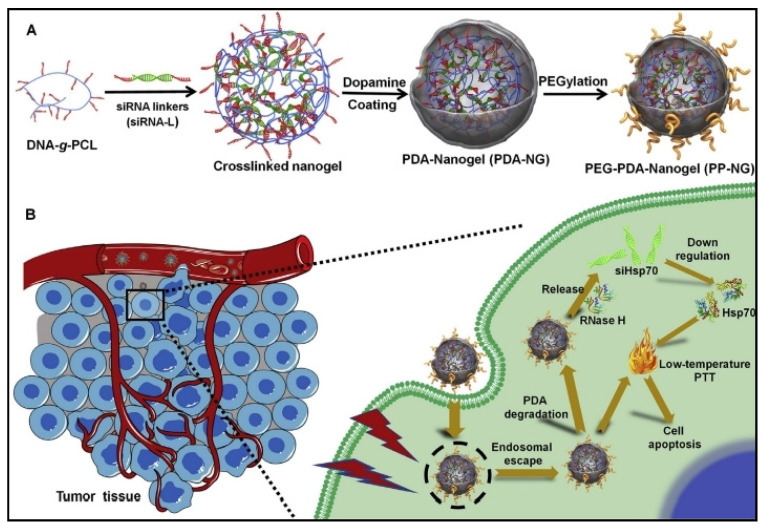
An example of a PDA-coated nucleic acid nanogel and its use in siRNA-mediated low-temperature PTT therapy in vivo. (**A**) The PDA-coated nucleic acid nanogel’s (PEG-PDA-Nanogel) synthetic process. (**B**) The mechanism of PEG-PDA-Nanogel-induced low-temperature photothermal treatment. Reproduced with permission from Ref [62]. Copyright 2020, Elsevier.

**Figure 4 cells-13-00908-f004:**
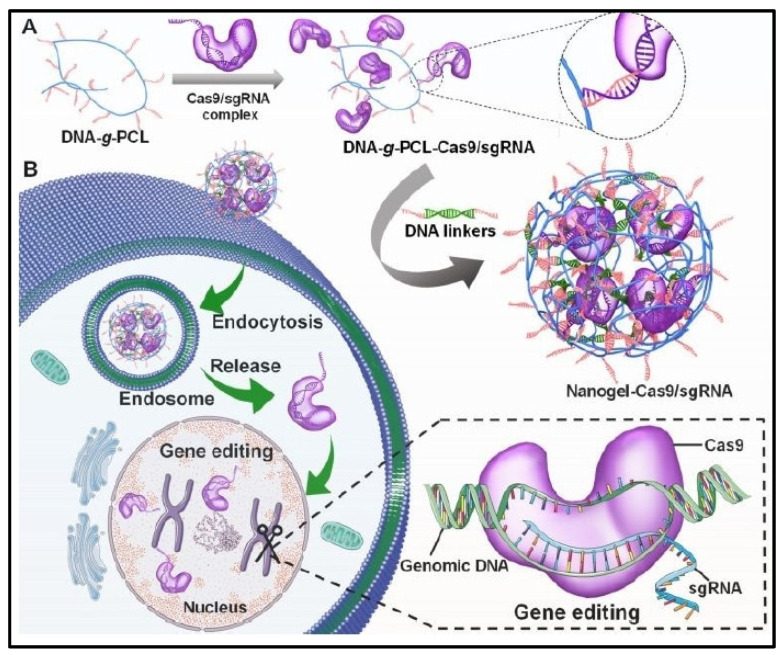
In vitro delivery of the Cas9/sgRNA-embedded nucleic acid nanogel. (**A**) The procedure for creating a nucleic acid nanogel with Cas9/sgRNA inserted. (**B**) The simultaneous delivery of the Cas9 and sgRNA embedded in the nanogel for in vitro gene editing. Reproduced with permission from Ref [71]. Copyright 2019, Royal Society of Chemistry.

**Figure 5 cells-13-00908-f005:**
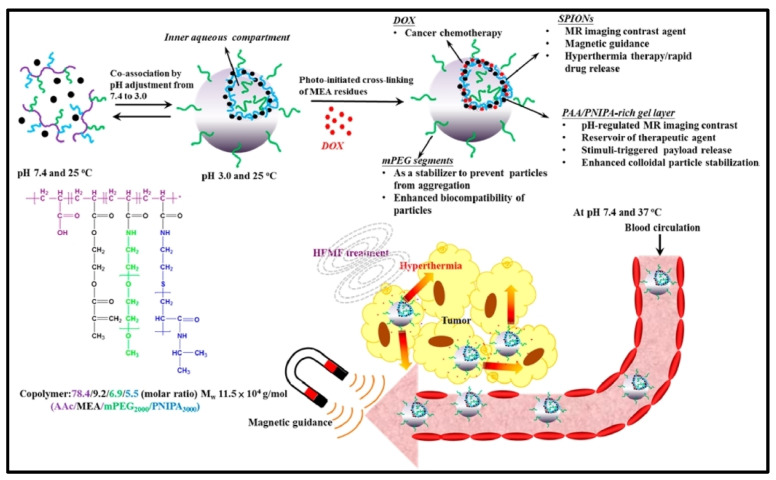
Hybrid nanogels can be used as multimodal theranostic vehicles. Schematic illustration of SPION including DOX-loaded hybrid nanogels with PEG and PNIPAM as grafts and acrylic acid and methacryloylethyl acrylate units as the backbone. Since the graft copolymer is pH- and temperature-sensitive, SPION was added to give MR imaging and magnetic guidance. Adapted with permission from Ref [81]. Copyright 2013, ACS Publications.

**Figure 6 cells-13-00908-f006:**
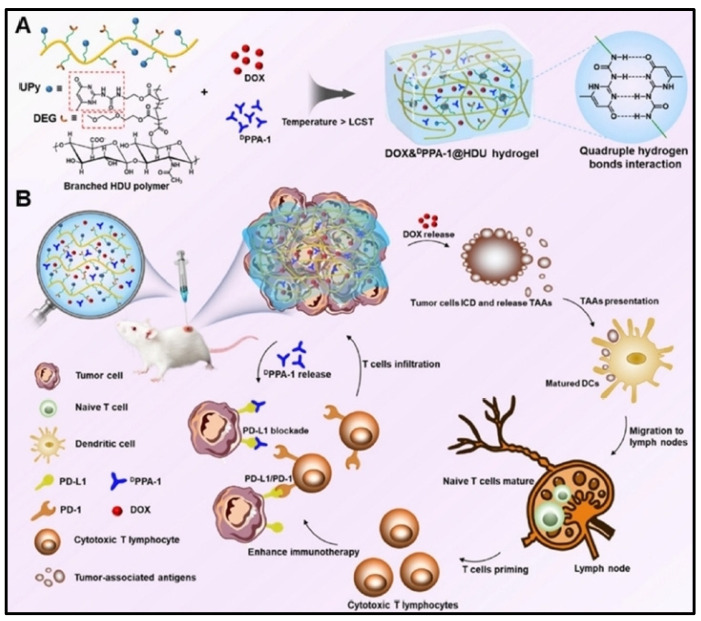
An injectable biomimetic thermoresponsive hydrogel with DPPA-1 and DOX is shown schematically. (**A**) The synthesis of a thermoresponsive hydrogel using DOX and DPPA-1. Under temperature-sensitive conditions, the DEG segment may create hydrophobic domains, and the UPy segment could create quadruple hydrogen bonds within the hydrophobic domains to create a hydrogel. (**B**) The DOX&DPPA-1@HDU hydrogel’s method of triggering immune responses. Reproduced with permission from Ref [83]. Copyright 2021, ACS Publications.

**Figure 7 cells-13-00908-f007:**
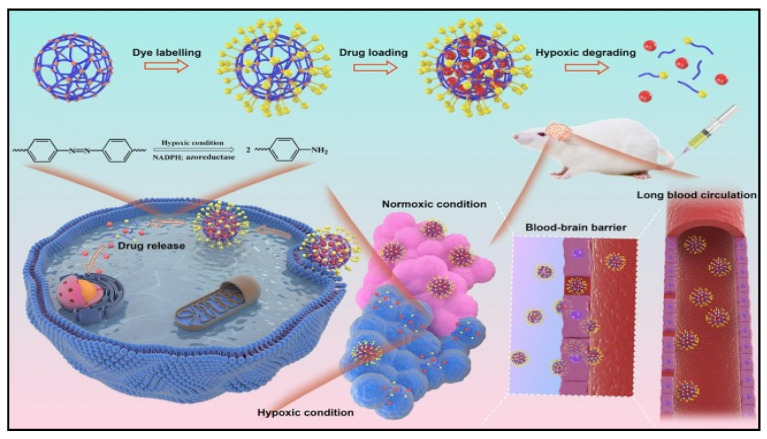
Diagrammatic representation of the hypoxia-controlled drug release and lengthy blood circulation of the poly(phosphorylcholine)-based (^H^PMPC) nanogel for tumor drug delivery. Reproduced with permission from Ref [91]. Copyright 2021, Elsevier.

**Table 1 cells-13-00908-t001:** Examples of hybrid nanogel drug delivery systems used for cancer therapy.

Form	Therapeutic Agents	Cancer Type	References
Nucleic acid nanogel	miR155	Glioblastoma	[54]
Graphene oxide hydrogel	mRNA/R848	Melanoma	[55]
PDA–nucleic acid nanogel	siRNA	Cervical cancer	[56]
Cisplatin nanogel	Doxorubicin	4T1 cells, melanoma	[57]
^125^I-GNR-RGDY hydrogel	Iodine-labeled RGDY	Breast cancer	[58]
siRNA/PPA-NG nanogel	PD-L1 siRNA	Melanoma	[59]
CRISPR/Cas9 nanogel	Cas9 and sgRNA	Hela cells	[65]
Tumor-editing hydrogel	IFN-β, TNF-α, and IL-12	Melanoma	[67]
Dual-stimulus nanogel	MR/ICG	Breast cancer	[74]
Immunomodulatory multidomain nanogel (iGel)	Gemcitabin, R837	Triple-negative breast cancer, TC1 cervical cancer cells	[75]
Biomimetic thermoresponsive hydrogel	Doxorubicin, PPA-1	CT26 colorectal cancer cells	[76]
TME responsive nanogel	Paclitaxel, IL-2	Melanoma	[77]
Hypoxia inducible hydrogel	Gelatin, ferulic acid	ECFC endothelial cells	[82]
Injectable hydrogel	Axitinib, PCSK9 inhibitor	Melanoma	[84]
^H^PMPC nanogel	doxorubicin, hypoxia crosslinkers	HepG2 and 293T cells	[85]

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
