# Peer review of "Hybrid Nanogel Drug Delivery Systems: Transforming the Tumor Microenvironment through Tumor Tissue Editing"

_cells, 2024, doi:10.3390/cells13110908_

Round 1

Reviewer 1 Report

Comments and Suggestions for Authors

In this review article, Theodora Katopodi et al. have summarized the application of hybrid nanogel drug delivery systems in tumor treatment. While the manuscript provides some useful information regarding the properties and applications of nanogels, it requires several significant improvements:

1. The title of the manuscript does not accurately reflect the main focus of the content. It suggests a concentration on tumor tissue editing, which is not the central theme discussed throughout the majority of the article.

 2. There are discrepancies between the content of some sections and their corresponding subtitles. For example, the section titled “Nanogels for Cancer Immunotherapy” includes references that are not directly related to the topic of cancer immunotherapy. Furthermore, the discussion in this section is somewhat superficial. A more detailed exploration of the design, functionality, mechanisms, and effectiveness of nanogels in cancer immunotherapy would enhance the quality of this section. Similar issues are noted in the “Injectable Reprogrammable Nanogels/Hydrogels” section, which also lacks depth and specificity related to the stated topic.

3. It would be beneficial for each section to include a table summarizing the functions, compositions, and publications related to nanogels. This would provide a clearer and more organized presentation of the information, facilitating better understanding and comparison of the different types of nanogels discussed.

Reviewer 2 Report

Comments and Suggestions for Authors

The review entitled “Hybrid nanogel drug delivery systems: transforming the tumor microenvironment by tumor tissue editing” illustrates the potential of nanogels as drug delivery systems to enhance the efficacy of tumor therapy.

The topic is very interesting and complex and presented in a complex way. The review should be written in a simpler and more usable way for the reader considering the complexity of the topic.

I suggest inserting paragraph numbering and changing the reference numbering (non-Roman).

Furthermore, in describing the role of hybrid nanogels, it should be better explained whether the studies are on in vitro or in vivo models in order to clarify in which phase of the experimentation we are.

I reconsider this review after major revision

Comments on the Quality of English Language

The English language required minor revision

Round 2

Reviewer 1 Report

Comments and Suggestions for Authors

The authors have addressed all my concerns. 

Reviewer 2 Report

Comments and Suggestions for Authors

The authors responded to the requests completely, improving the quality of the work. This review may be accepted for publication